# Glaucoma Progression after Delivery in Patients with Open-Angle Glaucoma Who Discontinued Glaucoma Medication during Pregnancy

**DOI:** 10.3390/jcm10102190

**Published:** 2021-05-19

**Authors:** Duri Seo, Taekjune Lee, Joo Yeon Kim, Gong Je Seong, Wungrak Choi, Hyong Won Bae, Chan Yun Kim

**Affiliations:** Department of Ophthalmology, Institute of Vision Research, Severance Hospital, Yonsei University College of Medicine, Seoul 03722, Korea; dubong7@daum.net (D.S.); angelua@naver.com (T.L.); drjykim@yuhs.ac (J.Y.K.); gjseong@yuhs.ac (G.J.S.); wungrakchoi@yuhs.ac (W.C.); baekwon@yuhs.ac (H.W.B.)

**Keywords:** medication, glaucoma, pregnancy, progression

## Abstract

In this retrospective study, clinical characteristics and glaucoma progression of open-angle glaucoma (OAG) patients who discontinued intraocular pressure (IOP)-lowering medication during pregnancy were investigated. Glaucoma progression was determined using either serial visual field tests or optic disc/retinal nerve fiber layer (RNFL) photographs. Age, number of previous pregnancies, diagnosis, average IOP, IOP fluctuation, visual field mean deviation, pattern standard deviation, and RNFL thickness were examined, and their association with glaucoma progression was determined using linear regression analysis. Among 67 eyes (37 patients), 19 eyes (28.4%) exhibited glaucoma progression 13.95 ± 2.42 months after delivery. The progression group showed significantly higher mean IOP than the nonprogression group in the first, second, and third trimesters (*p* = 0.02, 0.001, and 0.04, respectively). The average IOP in the second^,^ and third trimesters and IOP fluctuation during the entire pregnancy were significantly associated with glaucoma progression according to a univariate analysis (*p* = 0.04, 0.031, and 0.026, respectively). In conclusion, IOP elevation during pregnancy is associated with glaucoma progression after delivery in patients who had discontinued medication during pregnancy. Therefore, close monitoring of glaucoma is necessary, particularly if patients discontinue medication during pregnancy, and appropriate intervention should be considered in case of increased IOP.

## 1. Introduction

Although glaucoma is common among older people, it is far less common in women, especially during pregnancy. However, it is much more difficult for clinicians to manage them because clinicians are forced to weigh the balance between preventing further damage to the optic nerves and protecting the fetus. Despite the challenge, there exist only a few systematic studies that investigated glaucoma progression or the safety of intervention during pregnancy. Thus far, clinicians rely on a limited number of case series studies or personal experiences for the management of glaucoma patients during their pregnancy [1,2]. Understandably, a randomized comparative study to determine the safe drugs and the extent to which they are suitable in pregnant patients may be ethically difficult. For this reason, none of the commercially available glaucoma medications are considered Federal Food and Drug Administration safety category A. Most glaucoma medications, except for brimonidine and dipivefrin that are category B, fall into category C. In other words, there are no completely safe and readily available glaucoma medications for pregnant women [3,4]. According to one survey, 71% of patients continued to receive glaucoma medications during pregnancy; however, many ophthalmologists were uncertain about their glaucoma management strategies [5].

In an attempt to assess the course of glaucoma following pregnancy and develop a better strategy to manage pregnant patients, this study retrospectively identified factors associated with the progression of glaucoma after delivery in patients who discontinued glaucoma medication use during pregnancy.

## 2. Materials and Methods

### 2.1. Participants

This retrospective study adhered to the tenets of the Declaration of Helsinki and its protocol was approved by the Institutional Review Board of Yonsei University (4-2020-1470). The requirement for informed consent was waived because of the retrospective nature of the study. The medical records of patients who visited the glaucoma center in Severance Hospital between January 2005 and May 2020 were reviewed. Among them, patients meeting the following criteria were considered for analysis in this study: (1) female patients with preexisting open-angle glaucoma (OAG) who had discontinued glaucoma medication during pregnancy; and (2) a minimum of two years of follow-up with at least five high-quality optic disc photographs, red-free retinal nerve fiber layer (RNFL) photographs, optical coherence tomography (OCT) images, or reliable visual field (VF) testing results (fixation loss < 20%, false-positive errors < 15%, and false-negative errors < 15%). All participants underwent complete ophthalmic examinations, including best-corrected visual acuity, intraocular pressure (IOP) assessment using Goldmann applanation tonometer, autorefraction keratometry (RK-3; Canon USA, Inc., Lake Success, NY, USA), slit-lamp biomicroscopy, dilated fundus examination, optic disc photography (Carl Zeiss Meditec, Jena, Germany), red-free RNFL photography (Carl Zeiss Meditec), and spectral-domain OCT (Carl Zeiss Meditec). The VF test (Humphrey Field Analyzer II; Carl Zeiss Meditec) was also performed. These tests were repeated at intervals of 3–12 months, as needed.

A diagnosis of OAG was made when a patient showed a glaucomatous VF defect confirmed by reliable VF examinations and a glaucomatous optic disc (cup-to-disc ratio > 0.7; inter-eye cup asymmetry > 0.2; neuroretinal rim notching; focal thinning; disc hemorrhage; or vertical elongation of the optic cup), while gonioscopic examination showed an open angle.

Individuals with the following conditions were excluded: (1) secondary causes of glaucomatous optic neuropathy and (2) neurologic or systemic diseases that may potentially affect the VF.

### 2.2. Determination of Glaucoma Progression

Glaucoma progression after pregnancy was determined either by serial VF testing or optic disc/RNFL photographs. Standard automated perimetry was performed using the Swedish Interactive Threshold Algorithm (SITA) standard 24-2 program of the Humphrey Field Analyzer II. This study included only reliable VF tests (described earlier). To be included in this study, a minimum of five reliable VF test results from separate visits were required. The visual field was considered to have progressed if the slope between mean deviation (MD) and age was calculated by linear regression analysis to be negative and statistically significant (*p* < 0.05).

Glaucoma progression in the optic nerve and RNFL photographs were identified by two independent observers (DS and TL), and any disagreements were resolved by a third adjudicator (CYK). Signs that were considered as progressive optic disc changes include focal/diffuse narrowing/notching of the neuroretinal rim, increased cup-to-disc ratio, and changed in the adjacent vasculature by comparison of serial disc photography images. Increased width or depth of an existing RNFLdefect or appearance of a new defect was noted as a sign of progression.

### 2.3. Statistical Analyses

Statistical analysis was performed using SPSS version 22.0 (SPSS Inc, Chicago, IL, USA). Kruskal–Wallis test was used to evaluate IOP changes during pregnancy. An independent t-test or Mann–Whitney U test was used to compare continuous variables between the progression and nonprogression groups depending on data normality. Categorical variables were analyzed using Fisher’s exact test. Univariate and multivariate logistic regression models were applied to identify factors associated with glaucoma progression. Since both eyes of some of the patients were included in the study, ocular variables were not independent. To account for the inter-eye correlations, we used a generalized estimating Equation [6,7]. Power calculation was conducted using Power Analysis and Sample Size 11 for Windows software package (NCSS Inc, LLC, Kaysville, UT, USA).

## 3. Results

### Participant Characteristics

A total of 37 patients who elected to discontinue glaucoma medications during pregnancy were included in this study. Preconception counseling was provided to female glaucoma patients of childbearing age at Severance Hospital Glaucoma Center. These patients were also educated on the potential risk of glaucoma drugs to the fetus and the risk of optic nerve damage in case of discontinuation of the drug during pregnancy. Following confirmation of pregnancy, patients were given the choice of whether to continue the drug treatment. Medical records of patients who chose to discontinue the drug were reviewed for the study. 

The mean follow-up period of the participants was 45.67 ± 24.07 months after confirmation of pregnancy. Before pregnancy, their mean IOP was 14.21 ± 2.54 mmHg while using 1.55 ± 0.91 glaucoma medications. The mean IOP during the first, second, and third trimesters of pregnancy was 14.40 ± 2.64, 14.56 ± 2.79, and 15.65 ± 3.08 mmHg, respectively. All patients included in this study restarted their pre-pregnancy IOP-lowering medications within two months of delivery. The slight difference in the restarting point stemmed from the variations in the intervals between delivery and clinic visits after delivery. In this study, the postpartum IOP was defined as the IOP measured at least one month after reusing their drug. The mean IOP postpartum was 14.58 ± 2.58 mmHg with 0.04 ± 0.21 glaucoma medications. At 6 to 12 months after delivery, the mean IOP was 14.23 ± 2.84 mmHg with 0.73 ± 0.94 glaucoma medications (Table 1). The mean IOP postpartum was not statistically different from that at any other follow-up time points (*p* = 0.32, Figure 1). 

Progression was detected in a total of 19 eyes (28.4%). All 19 eyes showed progression in optic disc/RNFL photographs in form of increased RNFL widths. VF progression was also detected in four of them. The average time to progression detection was 13.95 ± 2.42 months after confirmation of pregnancy (Figure 2). No differences were observed with regard to age, the number of previous pregnancies, baseline IOP, pre-pregnancy MD, and pattern standard deviation as well as RNFL thickness between the progression and nonprogression groups. However, the progression group presented significantly higher IOP in the first, second, and third trimesters (*p* = 0.02, 0.001, and 0.04, respectively) (Table 2). When the peak IOP during pregnancy was compared to that before pregnancy, IOP was elevated by 15.62% during pregnancy in the progression group and by 6.93% in the nonprogression group. According to our power calculation, sample sizes of 19 and 48 achieve 75% power in detecting a difference in mean IOPs when there was a difference of 1.9 between the null hypothesis mean difference of 0.0 and the actual mean difference of −1.9 at the 0.05 significance level (alpha) using a two-sided Mann–Whitney–Wilcoxon Test.

In univariate logistic regression analysis, the second and third trimester IOP and IOP fluctuation during pregnancy were found to be significantly associated with glaucoma progression (*p* = 0.040, 0.031, and 0.026, respectively) (Table 3). However, there were no factors that were statistically significant when further analysis was conducted using multivariate regression models.

## 4. Discussion

One of the most challenging situations faced by glaucoma specialists is the management of the disease during pregnancy. Although the prevalence of glaucoma is not very high in women of childbearing age, the proportion of young adults with glaucoma is increasing [8]. Considering their longer life expectancy, the management of patients with glaucoma during pregnancy cannot be taken lightly.

Many physiological changes occur in various organs and tissues during pregnancy, including the eyes. Some physiological changes reported by previous studies include changes in IOP, corneal sensitivity, outflow facility, and temporary refractive changes [9]. IOP is believed to decrease during the entire course of pregnancy [10,11,12,13,14] in part due to a combination of increased uveoscleral outflow as a result of hormonal changes and decreased episcleral pressure secondary to a general decrease of venous pressure in the upper extremities and systemic metabolic acidosis [15]. Elevated levels of estrogen, progesterone, relaxin, and beta-human chorionic gonadotropin levels during pregnancy are negatively correlated with changes in IOP [11]. Consistent with the role of elevated IOP in most glaucoma, this study also showed that glaucoma progression after pregnancy is associated with IOP elevation during pregnancy. This study also showed that although the IOP tends to decrease during pregnancy, the IOP changes during pregnancy cannot be predicted in advance in any individual patient and that this temporary IOP reduction may not prevent glaucoma progression overall. Furthermore, it is possible that IOP in pregnant women may be underestimated. The physiological softening of a ligament in late pregnancy may extend to the cornea to reduce corneal rigidity, making applanation tonometry readings falsely low [16,17].

In the management of glaucoma during pregnancy, all glaucoma drugs pose a potential risk to the fetus [18]. Therefore, relevant information must be provided to the patient and the use of the drug must be discussed thoroughly. Since glaucoma drug use may not be safe for the fetus, many mothers with glaucoma choose to discontinue drug use during pregnancy despite the risk of glaucoma progression. However, there are very few studies on glaucoma progression during or after pregnancy or on the effect of drug discontinuation during pregnancy on subsequent glaucoma progression. One previous study, which evaluated glaucoma in eight pregnant patients, demonstrated that glaucoma progression was noted in one patient, while the IOP and VF tests were stable for the rest of the study population [19]. Another study on 28 eyes of 15 pregnant women with glaucoma reported that 10 (35.7%) of 28 eyes showed an increase in IOP or a progression of VF loss during pregnancy [20]. However, since these studies included glaucoma patients who discontinued and those who only reduced medication during pregnancy, IOP control was not uniform. Therefore, their value in a clinical setting is limited. In this study, none of the patients showed glaucoma progression during the course of pregnancy. Most of the patients included in the study were either normal-tension glaucoma or primary open-angle glaucoma patients with relatively low pretreatment IOP. Additionally, the severity of glaucoma was mild to moderate in most patients. However, glaucoma progression was found in 28.4% of patients who discontinued glaucoma medication during pregnancy 13.95 ± 2.42 months after delivery. Although our analysis lacks a control group for comprehensive comparisons, this number alone is high enough to receive further attention.

It has been reported that the IOP decrease during pregnancy lasts up to 2 months postpartum [21]. However, the IOP increased slightly in the glaucoma progression group in our study, from 14.79 ± 2.89 mmHg before pregnancy to 17.67 ± 1.56 mmHg in the third trimester. In the nonprogression group, the IOP was 14.44 ± 3.17 mmHg in the third trimester, showing no significant change from 13.99 ± 2.38 mmHg before pregnancy. While it is challenging to accurately establish the maximum IOP values that should prompt treatment from the results of this study, our study results suggest that elevated IOP during pregnancy is related to ensuing glaucoma progression after delivery. Given that the progression group showed the mean IOP elevation in the third trimester by 19.4%, compared with the pre-pregnancy IOP, we propose that it is likely necessary to consider the use of drugs for patients showing similar levels of IOP increase. If patients are reluctant to use glaucoma medication, laser or surgery may be a reasonable alternative. A case series of selective laser trabeculoplasty (SLT) found the procedure to be effective in eliminating the need for glaucoma medications before pregnancy [2]. SLT is also conceivable for patients during their pregnancy since it would impose minimum risks to the fetus and help avoid the addition of perioperative and postoperative medications at the same time. Choosing the surgical options to control IOP during pregnancy involves concerns with anesthesia and postoperative medications. The recent introduction of minimal invasive glaucoma surgery (MIGS) provides clinicians with an option that is simple and safe while eliminating risks associated with anesthesia. In addition, in the case of glaucoma patients who have a high risk of deterioration during pregnancy or are already at an advanced stage, it would be desirable to achieve a stable IOP through surgery before conception [2,22,23].

This study has some limitations. First, since this is a retrospective study, the timing of various examinations and duration of follow-up are different among the patient population. This difference may have limited the power of the analysis. Second, not all of the factors possibly associated with glaucoma progression were investigated. For instance, it is known that vascular risk factors such as migraine, hypertension, diabetes, anemia, and myopia increase the risks of progression, especially in NTG [24]. Since this study contains a significant portion of NTG patients, it is possible that factors other than IOP may have played an important role in the progression of glaucoma. The omission of analysis of such risk factors is admittedly a major drawback of this study. However, as aforementioned, because prospective studies on the use of glaucoma drugs in pregnant women are extremely difficult and can pose ethical issues and there are very few studies on the progression of patients with glaucoma during pregnancy or after delivery, the value of retrospective studies on this topic cannot be overstated. Third, the small number of participants limits the power of the study. Due to the inherent nature of the topic, it was very difficult to secure a larger number of eligible patients to increase its power. Although the power is less than 0.80, we believe it is close enough to be considered reliable evidence with which to guide clinical management of pregnant glaucoma patients. It is also possible that the absence of any statistically significant variables in multivariate analysis is due to the small number of patients showing progression. Fourth, the method of evaluating glaucoma progression was limited. VF deterioration was noted in only four eyes, and the progression of the damage was confirmed using the RNFL photograph in most cases. Some progression may have been missed, particularly in those glaucoma patients with no discernible localized RNFL defects or unclear damage boundaries. It was difficult in this study to evaluate the progression of glaucoma using optical coherence tomography (OCT), which is currently the most widely used form. This is because our data included patients who were examined with different types of OCT during the period between 2005 and 2020. Fifth, the patient population includes both NTG and high-tension glaucoma (HTG). Even though NTG and HTG are thought to exist at the two ends of the broad spectrum of a single disease, several significant differences have been reported. However, most of the HTG cases in this study may share more characteristics with NTG than otherwise known because our HTG cases showed relatively low IOP even when the drug was discontinued altogether. In addition, some myopic NTG cases are reported to show very slow glaucoma progression or none at all, but this study did not include analysis on the effect of refractive errors. Lastly, the patient population is a homogenous group comprised of a single ethnicity (Korean), from a country where normal-tension glaucoma is the most common form of glaucoma; therefore, caution should be exercised in generalizing our results. Since our study on such patients with only moderately high IOP showed glaucoma progression to occur in 28.4% when glaucoma medication is discontinued for 10 months, we believe that discontinuation of glaucoma medication during pregnancy should be decided much more carefully for cases of glaucoma with high IOP.

Nevertheless, to the best of our knowledge, this is the first of its kind to report postpartum glaucoma progression after discontinuing glaucoma medication completely during pregnancy, and we believe these results help understand the course of glaucoma progression during and after pregnancy.

## 5. Conclusions

In conclusion, IOP elevation during pregnancy is associated with subsequent glaucoma progression after delivery in pregnant patients who had discontinued medication. Therefore, close monitoring of glaucoma is necessary, particularly if patients discontinue glaucoma medication during pregnancy, and the use of medication or other therapeutic options such as laser or surgery should be considered in case of IOP elevation during pregnancy.

## Figures and Tables

**Figure 1 jcm-10-02190-f001:**
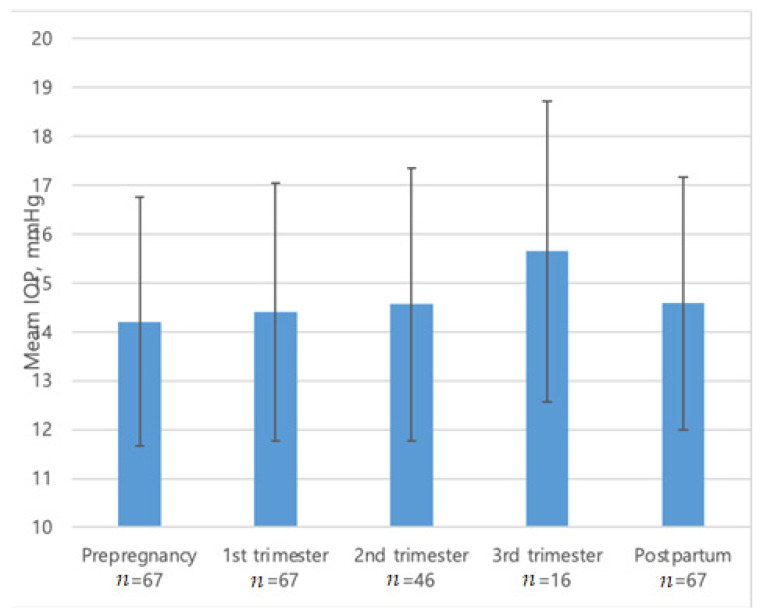
Mean intraocular pressure (IOP) of study subjects. There was no difference in IOP between the five follow-up periods by Kruskal–Wallis test (*p* = 0.32).

**Figure 2 jcm-10-02190-f002:**
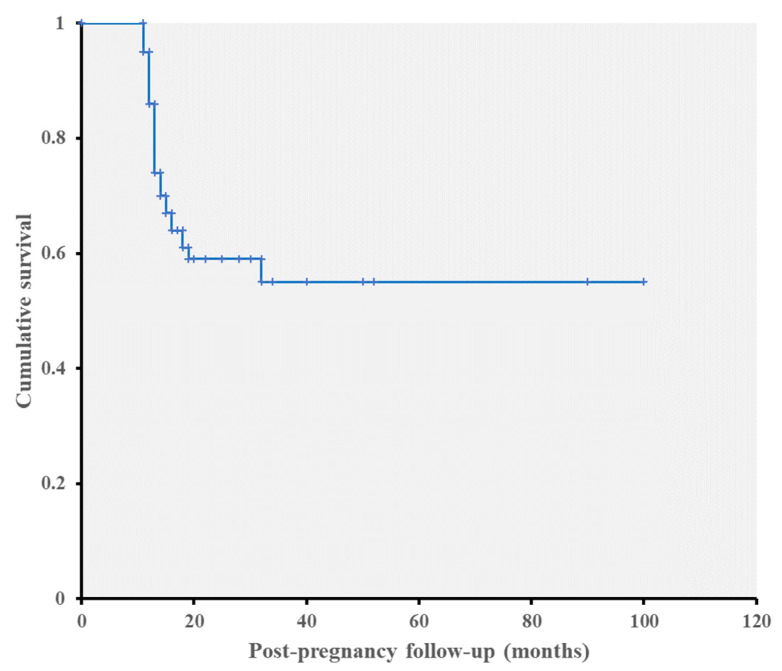
Kaplan–Meier cumulative survival analysis curve of glaucoma progression after pregnancy.

**Table 1 jcm-10-02190-t001:** Clinical characteristics of the study population.

	All Subjects (N = 37)
Age (yr)	32.40 ± 2.94
Number of eyes	67
Right eyes	34 (50.7%)
Left eyes	33 (49.3%)
Prior pregnancies	0.09 ± 0.29
Number of medications in pre-pregnancy	1.55 ± 0.91
Pre-pregnancy IOP (mmHg)	14.21 ± 2.54
Pre-pregnancy MD (dB)	−5.25 ± 5.11
Pre-pregnancy PSD (dB)	5.62 ± 4.43
Pre-pregnancy RNFL thickness (µm)	75.55 ± 10.18
Follow-up periods after pregnancy (months)	45.67 ± 24.07
Diagnosis	
Normal-tension glaucoma	44 (65.7%)
Primary open-angle glaucoma	23 (34.3%)

Values are presented as mean ± standard deviation for continuous variables, and number (percent) for categorical variables: IOP = intraocular pressure; MD = mean deviation; PSD = pattern standard deviation; RNFL = retinal nerve fiber layer; VF = visual field.

**Table 2 jcm-10-02190-t002:** Comparisons of the clinical characteristics between glaucoma progression and nonprogression groups during pregnancy without medication.

	Progression (N = 19)	Non-Progression (N = 48)	*p* Value
Age	32.21 ± 2.04	32.48 ± 3.25	0.97
Number of pre-pregnancy medications	0.16 ± 0.38	0.06 ± 0.25	0.22
Pre-pregnancy IOP (mmHg)	14.79 ± 2.89	13.99 ± 2.38	0.25
First trimester IOP (mmHg)	15.75 ± 3.31	13.86 ± 2.15	0.02
Second trimester IOP (mmHg)	16.50 ± 2.80	13.62 ± 2.28	0.001
Third trimester IOP (mmHg)	17.67 ± 1.56	14.44 ± 3.17	0.04
Postpartum IOP (mmHg)	15.48 ± 2.29	14.16 ± 2.61	0.06
MD (dB)	−4.52 ± 4.83	−5.56 ± 5.25	0.33
PSD (dB)	5.15 ± 4.05	5.82 ± 4.60	0.63
RNFL thickness (µm)			
Average	74.42 ± 9.73	76.02 ± 10.42	0.56
Temporal	64.68 ± 13.77	65.43 ± 16.04	0.86
Superior	94.53 ± 20.39	95.07 ± 22.28	0.93
Nasal	58.16 ± 11.01	60.54 ± 9.15	0.41
Inferior	81.00 ± 16.53	82.74 ± 22.52	0.78
Diagnosis			0.08
Normal-tension glaucoma	9	35	
Primary open-angle glaucoma	10	13	

Values are presented as mean ± standard deviation: IOP = intraocular pressure; MD = mean deviation; PSD = pattern standard deviation; RNFL = retinal nerve fiber layer.

**Table 3 jcm-10-02190-t003:** Risk factors for glaucoma progression by univariate logistic regression models.

Variable	Coefficient (95% CI)	*p* Value
Age	0.991 (0.819–1.198)	0.923
Prepreg. No	0.349 (0.043–2.833)	0.324
Diagnosis (NTG/HTG)	0.405 (0.120–1.373)	0.147
Number of medications before pregnancy	0.280 (0.056–1.411)	0.123
Progression before pregnancy	0.565 (0.242–1.318)	0.187
Pre-pregnancy IOP (mmHg)	1.103 (0.907–1.341)	0.328
First trimester IOP (mmHg)	1.229 (0.969–1.559)	0.089
Second trimester IOP (mmHg)	1.437 (1.017–2.031)	0.040
Third trimester IOP (mmHg)	1.755 (1.054–2.924)	0.031
IOP fluctuation in pregnancy (mmHg)	1.373 (1.039–1.814)	0.026
Postpartum IOP (mmHg)	1.088 (0.897–1.319)	0.393
MD (dB)	1.007 (0.942–1.076)	0.835
PSD (dB)	1.015 (0.923–1.115)	0.764
RNFL thickness		
Average	0.992 (0.977–1.008)	0.344
Temporal	1.003 (0.987–1.019)	0.721
Superior	0.995 (0.981–1.009)	0.496
Nasal	1.008 (0.964–1.054)	0.721
Inferior	0.994 (0.977–1.012)	0.521

IOP = intraocular pressure; MD = mean deviation; PSD = pattern standard deviation; RNFL = retinal nerve fiber layer.

## Data Availability

Data sharing is not applicable to this article.

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
