# Peer review of "Glaucoma Progression after Delivery in Patients with Open-Angle Glaucoma Who Discontinued Glaucoma Medication during Pregnancy"

_jcm, 2021, doi:10.3390/jcm10102190_

Round 1

Reviewer 1 Report

SENT April,3, 2021

RE:  Manuscript review JCM – 1175511

          The manuscript entitled “Glaucoma progression after delivery in patients with open-angle glaucoma who discontinued glaucoma medication during pregnancy” deals with the assessment of the risk factors for glaucoma progression in young females affected by primary open-angle glaucoma during and after pregnancy.

The study topic is quite interesting and it is of clinical relevance considering that the management of glaucoma during pregnancy could be extremely difficult, requiring a delicate balance between the potential risk for the fetus related to the anti-glaucomatous medications, and the need of preventing glaucoma progression of the mother. Only few studies dealing with the glaucoma progression or with the safety of anti-glaucoma medications during pregnancy are present in the literature.

The study under review, however, has many drawbacks, especially from a methodological point of view.

Several specific major issues need to be addressed, which include:

  1. Methods: the sample size of the population included in the present study seems to be definitely low (67 eyes of 37 patients) to provide results with sense from a statistical point of view. The authors should calculated the power of the study in discriminating between differences of 3 mmHg in intraocular pressure (that is known to be the test-retest variability of the Goldmann applanation tonometer used in the present study, and that is also the greatest difference found by the authors between progression and non-progression groups), and to increase the sample of eyes in order to provide a power >0.80, with an alfa error of 0.05;

  1. Methods: the population included in the present study seems to be definitely too heterogeneous to provide meaningful results. In particular, both cases of high-tension (HTG) (23 eyes) and normal-tension glaucoma (NTG) (44 eyes) were included. A controversy surrounding NTG is the question of whether it should be regarded as a disease within the spectrum of POAG or as a distinctive disease entity. NTG does have, indeed, several distinctive features compared with POAG: intraocular pressure-independent risk factors for development, characteristic patterns of structural and functional damage, and a unique disease course. For these reasons, several previous authors have regarded NTG and POAG as two different entities (Mastropasqua R, Fasanella V, Agnifili L, Fresina M, Di Staso S, Di Gregorio A, Marchini G,Ciancaglini M: Advance in the pathogenesis and treatment of normal-tension glaucoma; Prog Brain Res.2015;221:213-32. Killer HE, Pircher A; Normal tension glaucoma: review of current understanding and mechanisms of the pathogenesis. Eye (Lond).2018 May;32(5):924-930.  Kim KE, Park KH. Update on the Prevalence, Etiology, Diagnosis, and Monitoring of Normal-Tension Glaucoma. Asia Pac J Ophthalmol (Phila). 2016 Jan-Feb;5(1):23-31. Häntzschel J, Terai N, Sorgenfrei F, Haustein M, Pillunat K, Pillunat LE. Morphological and functional differences between normal-tension and high-tension glaucoma. Acta Ophthalmol. 2013 Aug;91(5):e386-91. Pruzan NL, Myers JS. Phenotypic differences in normal vs high tension glaucoma. J Neuroophthalmol 2015 Sep;35 Suppl 1:S4-7. Mursch-Edlmayr AS, Waser K, Podkowinski D, Bolz M. Differences in swept-source OCT angiography of the macular capillary network in high tension and normal tension glaucoma. Curr Eye Res. 2020 Feb 3:1-5. Li L et al.  Posterior displacement of the lamina cribrosa in normal-tension and high-tension glaucoma. Acta Ophthalmol. 2016 Sep;94(6).)  Considering that  the HTG and NTG are regarded as two different pathologies with different risk factors, clinical course and prognosis, the combination the these two types of glaucoma could introduce a confounding factor. I strongly suggest the authors to significantly increase the sample size and to consider HTG and NTG separately.

Furthermore, in order to avoid other possible bias, and considering that no mention about the refractive error of the population included in the study has been provided, it will be important to know the percentage of  patients having the myopic form of NTG that were included. As known, the myopic NTG is indeed regarded as a different clinical entity with peculiar characteristics (Han JC, Han SH, Park DY, Lee EJ, Kee C. Clinical Course and Risk Factors for Visual Field Progression in Normal-Tension glaucoma with myopia without glaucoma medications. Am J Ophthalmol. 2020 Jan;209:77-87).  

  1. Methods: several well known variables strictly related to glaucoma progression, especially in cases of NTG, have not been taken into consideration, which include:
  • Mean, range and maximum peak IOP during and after pregnancy (Nouri-Mahdavi K et al. Predictive factors for glaucomatous visual field progression in the advanced Glaucoma Intervention Study. Ophthalmology 2004; 111:1627-1635; Rao et al. Relationship between intraocular pressure and rate of visual field progression in treated glaucoma. J Glaucoma 2013;22:719-724);
  • Corneal hysteresis and its possible modifications during pregnancy (Park JH et al. Significance of corneal biomechanical properties in patients with progressive normal-tension glaucoma. Br J Ophthalmol 2015, 99:746-751);
  • Presence of optic disc hemorrhage (Ishida K et al. Disk hemorrhage is a significant negative prognostic factor in normal-tension glaucoma. Am j Ophthalmol 2000;129:707-714; De Moraes et al. Risk factors for visual field progression in treated glaucoma. Arch Ophthalmol 2011;129:562-568);
  • Presence and degree of myopia (Sakata et al. Contributing factors for progression of visual field loss in normal-tension glaucoma patients with medical treatment. J Glaucoma 2013;22:250-254);
  • Presence of tilted disc (Sung et al. Optic disc rotation as a clue for predicting visual field progression in myopic normal-tension glaucoma. Ophthalmology 2016;123:1484-1493);
  • Systemic blood pressure parameters (mean, fluctuation, minimum and maximum systolic and diastolic pressure, mean systolic and diastolic perfusion pressure), which are likely to change during pregnancy (Lee et al. Risk factors associated with structural progression in normal-tension glaucoma: intraocular pressure, systemic blood pressure, and myopia. IOVS 2020;61:1-10; Tham et al. Inter-relationship between ocular perfusion pressure, blood pressure, intraocular pressure profiles and primary open-angle glaucoma: the Singapore Epidemiology Eye Diseases Study. Br J Ophthalmol 2018; 102:1402-1406; Charlson et al. Nocturnal systemic hypotension increases the risk of glaucoma progression. Ophthalmology 2014;121:2004-2012.);
  • Use of systemic anti-hypertensive medications (Muskens et al. Systemic antihypertensive medication and incident open-angle glaucoma. Ophthalmology 2007;114:2221-2226);
  • Presence of obstructive nocturnal sleep apnea/hypopnea (Lin et al. Normal tension glaucoma in patients with obstructive sleep apnea/hypopnea syndrome. J Glaucoma 2011;20:553-558.);
  • Presence of blood rheology impairment (Cheng et al. The hemorheological mechanisms in normal tension glaucoma. Curr Eye Res 2011;36:647-653);
  • Presence of migraine (Ernest et al. An evidence-based review of prognostic factors for glaucomatous visual field progression. Ophthalmology 2013;120:512-519);
  • Presence of diabetes or glycemic intolerance, which also could be frequent during pregnancy (Newman-Casey et al. The relationship between components of metabolic syndrome and open-angle glaucoma. Ophthalmology 2011;118:1318-1326).
  1. Methods: As underlined by the same authors as a limitation of the study, the method use to evaluate glaucoma progression was limited. The determination of the glaucoma progression is a very challenging issue (Tanna AP. The challenge of detecting glaucoma progression. Ophthalomology 2017;124:S49-S50). The methods use in the present study (optic disc biomicroscopic evaluation and agreement amongst experts) seems to be extremely weak (Shah et al. Provider agreement in the assessment of glaucoma progression within a team model. J Glaucoma 2018;27:691-698);

Although the topic of this study is quite interesting, the manuscript is limiting in nature. In particular, number of participants, inclusion criteria and ocular and systemic risk factors investigated need to be seriously re-addressed.

Author Response

Replies to reviewer comments are organized as attached files.

Reviewer 2 Report

The authors in their paper assesed Glaucoma Progression After Delivery in Patients with OpenAngle Glaucoma Who Discontinued Glaucoma Medication During Pregnancy. This is very interesting paper and I read it with a pleasure, however some issues need to be cleared, as they are imprecise.

Abstract:

"disc/retinal nerve fiber layer (RNFL) photographs-?" It is not clear why and when did you used photographs, since you have also OCT device. Please clarify (in method section) how many patiens had photo, how many OCT? If all of the patients had OCT it is more precise method to access the progression than photo.

"medication should be considered in case of increased IOP." - I totally disagree since other option are available (SLT/MIGS surgeries)

Introduction

"The results of this study would help increase the
knowledge and understanding of drug use during pregnancy in patients with glaucoma." - I think you missed the point. The result of the study helps physician to assess the risk of glaucoma progression and to take right decision regarding to the patients prognosis. Please rephrease that.

Methods: Did the subjects take their medictaions after delivery again? If yes, when they started? If not, why?

Please explain exactly the issue with the OCT and RNFL photos (as mentioned above)

Discussion:

 "SLT also can be conceivable during pregnancy as it would
impose the minimum possible risk to the fetus, and would not require the addition of perioperative and postoperative medications" Thats perfectly true. Please add the sentence, that other methods such as minimal invasive glaucoma surgery can be consider during pregnacy, or before conception to secure optic nerve disc and add following references:  PMID: 32021062; and PMID: 32676205; 

Author Response

(The authors gave the same response as above.)

Reviewer 3 Report

It is quite well written although retrospective. However, the topic is very important and possesses potential for future citing.

2 main concerns need to be elucidated:

  1. Lack of actual results of VF progresssion, it should be added. Sole statement of progression in 19% of women is not enough.
  2. Though authors admit that limitation of the study is that Korean women are NTG almost always but in our clinical practice most problematic patients are HTG patients (80% of the pregnant women). It severely reduces the soundness of the study as the cessation of drops for 9 months is not the same for NTG vs HTG. It should be stressed better and perhaps even reflected  in the title.

Author Response

(The authors gave the same response as above.)

Reviewer 4 Report

This manuscript is quite interesting; however, some improvements are needed, in particular the discussion paragraph. 
The mean IOP was 14.21 ± 2.54 mmHg when using the drug before pregnancy. Is it related to this manuscript?
It has been reported that the lowered IOP during pregnancy lasts up to 2 months postpartum [21].
However, the IOP increased slightly in the glaucoma progression group and in- creased to 17.67 ± 1.56 mmHg in the third trimester. In the group where the progression of glaucoma was not detected, the IOP was 14.44 ± 3.17 mmHg in the third trimester, showing no significant change.
Please clarify these sentences. What was the mean?
Moreover, discuss more deeply the study results, comparing with the other two cited studies.

Author Response

(The authors gave the same response as above.)

Round 2

Reviewer 1 Report

RE:  Manuscript review JCM – 1175511R1

          The manuscript entitled “Glaucoma progression after delivery in patients with open-angle glaucoma who discontinued glaucoma medication during pregnancy” deals with the assessment of the risk factors for glaucoma progression in young females affected by primary open-angle glaucoma during and after pregnancy.

The study topic is quite interesting and it is of clinical relevance considering that the management of glaucoma during pregnancy could be extremely difficult, requiring a delicate balance between the potential risk for the fetus related to the anti-glaucomatous medications, and the need of preventing glaucoma progression of the mother. Only few studies dealing with the glaucoma progression or with the safety of anti-glaucoma medications during pregnancy are present in the literature.

The study under review, however, has many drawbacks, especially from a methodological point of view.

Reply to comment 1: OK

Reply to comment 2: I still believe that the study population was too heterogeneous and that a precise definition of the glaucoma type, that it extremely important in order to investigate risk factors and prognosis, is lacking. In particular, if the majority of patients were affected by a NTG (as stated by the authors: “most of the HTG patients in this study who can stop glaucoma medications during pregnancy can have some similar characteristics to NTG because IOP is not that high when they stop”), the only evaluation of the IOP measurements as risk factor for glaucoma progression appears definitely reductive;

Reply to comment 3: I still believe that too many variables strictly related to glaucoma progression, especially in cases of NTG where the IOP values seem to play a secondary role, have not been taken into considerations;

Reply to comments 4: I still believe that the method used in the present study in order to evaluate glaucoma progression was definitely weak, so that results could be controversial;

Furthermore, a comparison with a control group is lacking. What about glaucoma progression after delivery in patients with open-angle glaucoma who not discontinued glaucoma medication during pregnancy? What about glaucoma progression percentage in a similar number of participants with same age range and glaucoma type during similar follow-up duration?

Author Response

The author's reply was attached as a separate file.

Reviewer 1,

The manuscript entitled “Glaucoma progression after delivery in patients with open-angle glaucoma who discontinued glaucoma medication during pregnancy” deals with the assessment of the risk factors for glaucoma progression in young females affected by primary open-angle glaucoma during and after pregnancy.

The study topic is quite interesting and it is of clinical relevance considering that the management of glaucoma during pregnancy could be extremely difficult, requiring a delicate balance between the potential risk for the fetus related to the anti-glaucomatous medications, and the need of preventing glaucoma progression of the mother. Only few studies dealing with the glaucoma progression or with the safety of anti-glaucoma medications during pregnancy are present in the literature.

The study under review, however, has many drawbacks, especially from a methodological point of view.

Reply to comment 1: OK

Reply to comment 2: I still believe that the study population was too heterogeneous and that a precise definition of the glaucoma type, that it extremely important in order to investigate risk factors and prognosis, is lacking. In particular, if the majority of patients were affected by a NTG (as stated by the authors: “most of the HTG patients in this study who can stop glaucoma medications during pregnancy can have some similar characteristics to NTG because IOP is not that high when they stop”), the only evaluation of the IOP measurements as risk factor for glaucoma progression appears definitely reductive;

ANSWER:

Thanks again for your comments and advice. I agree with the reviewer's comments. The fact that the subjects of this study are heterogeneous is considered the limitation of this study. This point was also described in line 25 on page 10 of the discussion. However, despite such shortcomings, there are studies that include both NTG and HTG, such as EMGT. Also, in this study, there was no statistically significant difference in the frequency of NTG and HTG in the group with and without glaucoma progression after pregnancy (Table 2). In addition, since this study is a retrospective study and the number of subjects is small, there are limitations in analyzing other factors that influence glaucoma progression other than IOP. These facts are described again with more emphasis than before as follows.

In page 9, line 41,

It is known that various vascular risk factors such as migraine, hypertension, diabetes, anemia, and myopia can act as risk factors for glaucoma progression, especially in NTG [25]. Since this study contains more NTG patients, it is possible that various factors other than IOP may play an important role in the progression of glaucoma. The failure to analyze these points is considered to be a major drawback of this study.

Reference:

  1. Ernest PJ, Schouten JS, Beckers HJ, Hendrikse F, Prins MH, Webers CA: An evidence-based review of prognostic factors for glaucomatous visual field progression. Ophthalmol 2013, 120(3):512-519.

In page 10, line 25,

Fourth, the patient in this study is heterogeneous because both normal-tension glaucoma (NTG) and high-tension glaucoma (HTG) are mixed. NTG and HTG are thought to exist at both ends of the broad spectrum of a single disease, but several different characteristics are also reported.

Reply to comment 3: I still believe that too many variables strictly related to glaucoma progression, especially in cases of NTG where the IOP values seem to play a secondary role, have not been taken into considerations;

ANSWER:

Thank you for your comment. As mentioned above, the limitations of this study are described as the above by emphasizing the points on this part once again.

In page 9, line 41,

It is known that various vascular risk factors such as migraine, hypertension, diabetes, anemia, and myopia can act as risk factors for glaucoma progression, especially in NTG [25]. Since this study contains more NTG patients, it is possible that various factors other than IOP may play an important role in the progression of glaucoma. The failure to analyze these points is considered to be a major drawback of this study.

Reference:

  1. Ernest PJ, Schouten JS, Beckers HJ, Hendrikse F, Prins MH, Webers CA: An evidence-based review of prognostic factors for glaucomatous visual field progression. Ophthalmol 2013, 120(3):512-519.

Reply to comments 4: I still believe that the method used in the present study in order to evaluate glaucoma progression was definitely weak, so that results could be controversial;

ANSWER:

I agree with the reviewer's comments. Due to the limitations of retrospective case series study, it was not possible to use the same method such as GPA, a glaucoma progression detection method used in prospective study. This point was also described in line 16 on page 10 of the discussion.

In page 10, line 16,

Third, the method of evaluating glaucoma progression was limited. There were few deteriorations in the VF examination except four eyes, and the progression of the damage was confirmed using the RNFL photograph in most cases. Therefore, detection of progression may be overlooked in glaucoma patients with no localized RNFL defects or unclear damage boundaries. In addition, it was not possible to evaluate the progression of glaucoma using optical coherence tomography (OCT), which is widely used recently. Because the patient data from 2005 were used, the number of patients measured multiple times with the same type of OCT was too small. The actual deterioration was possibly not detected or detection was late owing to the limitations of retrospective studies.

Furthermore, a comparison with a control group is lacking. What about glaucoma progression after delivery in patients with open-angle glaucoma who not discontinued glaucoma medication during pregnancy? What about glaucoma progression percentage in a similar number of participants with same age range and glaucoma type during similar follow-up duration?

ANSWER:

Thank you for your comment. I agree with your opinion. Comparing glaucoma patients who stopped medication with glaucoma patients who continued to use glaucoma eye drops during pregnancy might yield stronger and more meaningful results. However, in my clinics, there were not many glaucoma patients who continue to use drug during pregnancy, and the duration of eye drops use during pregnancy was very different, and the number of eye drops was frequently not the same as before pregnancy. Therefore, pregnant glaucoma patients who did not completely stop their medication during pregnancy were so heterogeneous that it seemed not to be appropriate to set them as comparison group. Thus, the second best option was to take the form of a retrospective observational case series study. However, few studies have been conducted on the effects of discontinuation of glaucoma medication during pregnancy on later glaucoma progression because there are not many glaucoma patients who have gestational periods. In this regard, I think the results of this study are worthwhile despite various limitations.

Reviewer 2 Report

Thank you for constructive discussion about glaucoma during pregnacy. I feel satisfied with authors answers and corrections. 

Author Response

Thank you for constructive discussion about glaucoma during pregnacy. I feel satisfied with authors answers and corrections. 

ANSWER:

Thank you again for your comments and advice.

Round 3

Reviewer 1 Report

SENT May,4, 2021

RE:  Manuscript review JCM – 1175511R2

            The manuscript entitled “Glaucoma progression after delivery in patients with open-angle glaucoma who discontinued glaucoma medication during pregnancy” deals with the assessment of the risk factors for glaucoma progression in young females affected by primary open-angle glaucoma during and after pregnancy.

I still believe that the limitations of the present study (retrospective nature of the study; study population too small and too heterogeneous; lack of a control group; several variables related to glaucoma progression not taken into considerations; weak method for the evaluation of glaucoma progression) definitely reduce the meaning of the results and discussion.

Anyway, I am only one of the Reviewers of this manuscript, and my opinion is certainly limited by my own “forma mentis”  (mental structure).

Author Response

I have attached the answer to the review as a file.
